# Effect of Lumbar Spinal Stenosis on Treatment of Osteoporosis: Comparison of Three Oral Bisphosphonate Therapies

**DOI:** 10.3390/jcm12052027

**Published:** 2023-03-03

**Authors:** Hyung-Youl Park, Ki-Won Kim, Ji-Hyun Ryu, Geon-U Kim, Ho-Young Jung, Youn-Sung Jung, Jun-Seok Lee

**Affiliations:** 1Department of Orthopaedic Surgery, Eunpyeong St. Mary’s Hospital, College of Medicine, The Catholic University of Korea, Seoul 03312, Republic of Korea; 2Department of Orthopaedic Surgery, Yeouido St. Mary’s Hospital, College of Medicine, The Catholic University of Korea, Seoul 07345, Republic of Korea

**Keywords:** lumbar spinal stenosis, osteoporosis, bisphosphonate, bone mineral density

## Abstract

(1) Background: Lumbar spinal stenosis (LSS) causes uncomfortable neuropathic symptoms, which can negatively affect osteoporosis. The aim of this study was to investigate the effect of LSS on bone mineral density (BMD) in patients treated with one of three oral bisphosphonates (ibandronate, alendronate and risedronate) for initially diagnosed osteoporosis. (2) Methods: We included 346 patients treated with oral bisphosphonates for three years. We compared annual BMD T-scores and BMD increases between the two groups according to symptomatic LSS. The therapeutic efficacies of the three oral bisphosphonates in each group were also evaluated. (3) Results: Annual and total increases in BMD were significantly greater in group I (osteoporosis) compared to group II (osteoporosis + LSS). The total increase in BMD for three years was significantly greater in the ibandronate and alendronate subgroups than that in the risedronate subgroup (0.49 vs. 0.45 vs. 0.25, *p* < 0.001). Ibandronate showed a significantly greater increase in BMD than that of risedronate in group II (0.36 vs. 0.13, *p* = 0.018). (4) Conclusions: Symptomatic LSS may interfere with the increase in BMD. Ibandronate and alendronate were more effective in treating osteoporosis than risedronate. In particular, ibandronate was more effective than risedronate in patients with both osteoporosis and LSS.

## 1. Introduction

Osteoporosis is a systemic skeletal disorder characterized by fragile bones and more susceptibility to fractures. Although osteoporosis can occur in men, older women are at the highest risk due to low estrogen levels after menopause [1]. Most women are prescribed a bisphosphonate (BP) such as alendronate, risedronate, ibandronate or zoledronate as the first treatment option for osteoporosis [2,3]. As the elderly population increases, the prevalence of osteoporosis continues to increase. In 2010, the estimated prevalence of adults aged 50 or older was 10.3% (10.2 million) in the United States [4]. Osteoporosis and related complications can have serious economic effects on society and on individual quality of life. The burden of disease should be reduced by raising awareness of osteoporosis [5].

Similar to the high prevalence of osteoporosis in elderly patients, lumbar spinal stenosis (LSS) is also mainly caused by degeneration in the facet joints, intervertebral discs and ligaments narrowing the spinal canal and compressing the nerve roots [6]. LSS is a common and disabling cause of lower back and leg pain in elderly people. LSS affects an estimated 103 million people worldwide and 11% of the elderly in the United States [7]. LSS can cause various neurological symptoms, as follows: (1) leg or buttock discomfort aggravated when walking; (2) symptom relief while bending forward; (3) low back pain associated with such movement, including leaning forward to relieve symptoms; (4) motor or sensory changes when walking; and (5) weakness of the lower limbs [8]. These neurological symptoms of LSS decrease the physical activities of patients [9]. LSS can negatively affect the treatment of osteoporosis due to reduced daily activities.

Although the effect of LSS on osteoporosis has not been reported, our previous study revealed that symptomatic LSS may inhibit an increase in BMD in patients receiving ibandronate for initially diagnosed osteoporosis [10]. In the multivariate analysis, only LSS with symptoms was a risk factor in the inhibiting persistent increase in BMD (odds ratio = 0.316). The purpose of this study was to extend the results and evaluate the therapeutic efficacies of BPs in patients treated with one of three oral BPs (ibandronate, alendronate and risedronate). We aimed to evaluate the effect of LSS on BMD and to compare the increases in BMD in patients treated with three oral BPs for initially diagnosed osteoporosis.

## 2. Materials and Methods

### 2.1. Study Population

We retrospectively reviewed data from 1521 consecutive female patients who were initially diagnosed with postmenopausal osteoporosis in a single institution from January 2008 to December 2018. Clinical parameters of age, gender, BMD and body mass index (BMI) were evaluated using patients’ electronic medical records.

The inclusion criteria were (1) an annual BMD evaluation after the initial BMD evaluation that initially confirmed osteoporosis for three years (four BMD evaluations) and (2) oral bisphosphonate treatments (ibandronate, alendronate and risedronate) for three years. The dosages were 150 mg of ibandronate taken once monthly, 70 mg of alendronate taken once weekly or 35 mg of risedronate taken once weekly. All patients received vitamin D (800 international units/day) and calcium (600 mg/day) as an adjuvant therapy.

The exclusion criteria were (1) the continuous use of steroid hormones for the treatment of diseases, (2) a previous surgery or fractures of the lower extremities or spine, (3) a severe medical condition or malignant tumor that reduces physical activity and (4) a gait disorder due to causes other than LSS. After exclusion, 346 patients remained for analysis in this study. The study protocol was approved by the institutional review board (PC18RESE0034), ensuring compliance with the ethical guidelines of the Declaration of Helsinki 1975. Because of the retrospective study design, the need to obtain informed patient consent was waived by the institutional review board.

### 2.2. Diagnosis of Lumbar Spinal Stenosis

At the time of diagnosis of osteoporosis, the patients were divided into two groups according to symptomatic LSS. Among the patients who received oral BP medication for osteoporosis, those not complaining about LSS symptoms were included in group I (osteoporosis). Group II comprised patients with osteoporosis and symptomatic LSS (osteoporosis + LSS) (Figure 1). The inclusion criteria for symptomatic LSS were neurological symptoms consistent with central lumbar stenosis in magnetic resonance imaging (MRI) [11]. According to Lee’s MRI-based classification for central stenosis, patients with grade-two or three stenoses were included [12]. Patients with central lumbar stenosis, combined with concomitant lateral recess stenosis or foraminal stenosis in MRI, were also included [13,14].

The neurological symptoms of LSS included intermittent claudication and pain in the lower back, buttocks and/or posterior thighs that worsened with standing and walking and improved with sitting or leaning forward [6].

All patients were treated in a conservative manner with medications and epidural steroid injections for symptomatic LSS [15]. Steroid injections were performed only three or fewer times using 5 mg of dexamethasone. Patients undergoing spinal surgery due to worsening stenosis or failure of conservative treatment during the follow-up period were excluded from this study [16].

### 2.3. Measurement of Bone Mineral Density

Osteoporosis was diagnosed through dual-energy X-ray absorptiometry (DXA) (Lunar Prodigy; GE Healthcare Bio-Sciences Corp., Piscataway, NJ, USA). A total of four BMD T-score values were assessed initially and at 1-, 2-, and 3-year follow-ups. A BMD T-score ≤ −2.5 in the lumbar spine (total) or femur (neck or total) was defined as osteoporosis, and BMDs during follow-ups were measured in the same body region in which the initial osteoporosis had been diagnosed [17]. Discordant BMD values of the spine with aortic calcification and lumbar spondylosis were excluded. Discordant BMD values of the spine were defined as a discrepancy of >1 standard deviation in the T-score between adjacent vertebrae [18].

An annual increase in BMD for each group was measured as the difference in average BMD between two consecutive years (annual increase in BMD = index year BMD—previous year’s BMD). The total increase in BMD over three years was measured as the difference between the initial average BMD and the average three-year BMD (total increase in BMD = BMD at three-year follow-up—initial BMD). In this study, the changes in BMD were evaluated using T-scores.

### 2.4. Statistical Analyses

Perioperative continuous variables, presented as means and standard deviations, were compared between group I and group II using Student’s *t*-test, and categorical variables were compared using Fisher’s exact test. Analyses of variance with post hoc analysis using Bonferroni’s method were used to compare the parameters among the three oral bisphosphonates (ibandronate, alendronate and risedronate) in each group. Statistical analyses were performed using SPSS software (IBM SPSS Statistics for Windows, Version 24.0, IBM Corp., Armonk, NY, USA) with a level of statistical significance of 0.05.

## 3. Results

### 3.1. Comparisons between Groups I and II

In group I, 178 female patients with an average age of 74.8 ± 7.9 years were included; 168 female patients with an average age of 74.6 ± 8.3 years were included in group II. The BMIs and compositions of the three oral BPs were similar between the two groups (all *p* > 0.05). Although the initial BMD T-score and BMD T-score at the one-year follow-up were similar, those at the two-year and three-year visits were significantly greater in group I (−3.17 ± 0.66 vs. −3.36 ± 0.61, *p* = 0.006 at two years; −3.09 ± 0.49 vs. −3.31 ± 0.62, *p* < 0.001 at three years). The annual increase in BMD was significantly greater in group I than that in group II (0.29 ± 0.32 vs. 0.19 ± 0.33 at one year; 0.13 ± 0.29 vs. 0.06 ± 0.36 at two years; 0.11 ± 0.28 vs. 0.04 ± 0.33 at three years, all *p* < 0.05). Moreover, the total increase in BMD in group I over three years was significantly greater than that in group II (0.53 ± 0.47 vs. 0.28 ± 0.45, *p* < 0.001) (Table 1, Figure 2).

### 3.2. Comparison of the Three Oral BPs

Among the total 346 patients treated with oral bisphosphonates for three years, 142 used ibandronate, and 104 and 100 female patients were treated with alendronate and risedronate, respectively. The mean ages and BMIs were similar among the three BPs. The initial BMD T-score was significantly greater in the ibandronate group than that in the risedronate group (−3.67 ± 0.54 vs. −3.48 ± 0.57, *p* = 0.025 with a post hoc test). Increases in BMD at one year in the ibandronate and alendronate groups were significantly greater than those in the risedronate group (0.29 ± 0.31 vs. 0.29 ± 0.33 vs. 0.12 ± 0.33, *p* < 0.001). In a post hoc analysis, both ibandronate and alendronate groups had greater BMD increases at one year than those of the risedronate group (all *p* < 0.05). The total increase in BMD over three years in the ibandronate and alendronate groups was also significantly greater than that in the risedronate group (0.49 ± 0.46 vs. 0.45 ± 0.51 vs. 0.25 ± 0.43, *p* < 0.001). In a post hoc analysis, both ibandronate and alendronate groups had greater BMD increases over three years than that in the risedronate group (all *p* < 0.05) (Table 2).

### 3.3. Comparison of the Three Oral BPs within Each Group

The annual BMD T-scores and changes in BMD based on the type of BP in both LSS with or without osteoporosis groups are presented in Table 3. In group I, 76 patients used ibandronate, and 49 and 53 patients were treated with alendronate and risedronate, respectively. The mean ages and BMIs were similar among the three types of BP. The initial and annual BMD T-scores were not significantly different among the groups (all *p* > 0.05). However, BMD increases at one year in the ibandronate and alendronate groups were significantly greater than those in the risedronate group (0.34 ± 0.28 vs. 0.34 ± 0.32 vs. 0.16 ± 0.35, *p* = 0.002). The total increase in BMD over three years in the ibandronate and alendronate groups was significantly greater than that in the risedronate group (0.59 ± 0.40 vs. 0.62 ± 0.53 vs. 0.35 ± 0.46, *p* = 0.003). In a post hoc analysis, BMD increases at one year and over three years in both the ibandronate and alendronate groups were significantly greater than those in the risedronate groups (all *p* < 0.05) (Figure 3a).

In group II, 66 patients used ibandronate, and 55 and 47 patients were treated with alendronate and risedronate, respectively. The mean ages and BMIs were similar among the three types of BP. The initial and annual BMD T-scores were not significantly different among the groups (all *p* > 0.05). However, BMD increases at one year in the ibandronate and alendronate groups were significantly greater than those in the risedronate group (0.23 ± 0.33 vs. 0.24 ± 0.33 vs. 0.07 ± 0.31, *p* = 0.017). The total increase in BMD over three years was significantly greater in the ibandronate group than that in the risedronate group (0.36 ± 0.49 vs. 0.13 ± 0.38, *p* = 0.018 at post hoc test). Although the total increase in BMD in the alendronate was greater than that in the risedronate group, it was statistically insignificant (0.30 ± 0.45 vs. 0.13 ± 0.38, *p* > 0.05 at post hoc test) (Figure 3b).

## 4. Discussion

As both osteoporosis and LSS occur in elderly patients, the number of comorbid cases has been increasing. In a cross-sectional study, Lee et al. reported that osteoporosis or osteopenia was found in 79.2% of postmenopausal patients complaining of symptomatic LSS [19]. LSS causes various neuropathic symptoms and decreases the patient’s physical activity [9,20]. In our previous study, we demonstrated that BMD increases in only the osteoporosis group were greater than those of both the osteoporosis and LSS group for initially diagnosed osteoporosis treated with ibandronate [10]. In addition, only symptomatic LSS was a risk factor in the inhibiting persistent increase in BMD in the multivariate analysis. The results of the present study using three BPs are consistent with our previous study [10]. The annual increases in BMD were significantly greater in group I than those in group II (0.29 vs. 0.19 at one year; 0.13 vs. 0.06 at two years; 0.11 vs. 0.04 at three years). The total increase in BMD over three years was also significantly greater in group I (0.53 vs. 0.28). We suggested the mechanisms in a previous study. LSS causes neurologic claudication and decreases physical activity. Physical inactivity or walking difficulty due to claudication may be related to decreased BMD [21]. Physical activities can induce maintenance or increase BMD through a physical axial load. Therefore, active treatments for LSS should be undertaken to increase physical activity and BMD in patients with osteoporosis and LSS.

In this study, the therapeutic efficacies of three oral BPs (ibandronate, alendronate and risedronate) were compared. Our results reveal that ibandronate and alendronate had similar efficacy for newly diagnosed osteoporosis. This result is consistent with previous studies. In the results of the MOTION study, Miller et al. reported that monthly 150 mg ibandronate and weekly 70 mg alendronate increased 12-month BMD changes by an average of 2.9% and 3.0% in the total hip and 5.1% and 5.8% in the lumbar spine, respectively [22]. In their study, the clinical efficacy of ibandronate was comparable to that of alendronate with similar BMD increases in both the total hip and lumbar spine after 12 months. Moreover, a meta-analysis of randomized controlled trials for postmenopausal osteoporosis demonstrated that monthly ibandronate is equivalent to weekly alendronate, showing BMD increases in both the total hip and lumbar spine. However, monthly ibandronate is strongly preferred over weekly alendronate because of its convenience [23]. Cooper et al. also demonstrated the increased persistence of treatment in patients taking monthly ibandronate with patient support compared with weekly alendronate [24]. Increased compliance with bisphosphonate can be expected to improve the treatment of osteoporosis.

Comparing the three oral BPs, the total increase in BMD over three years in the ibandronate and alendronate groups was greater than that in the risedronate group (0.49 ± 0.46 vs. 0.45 ± 0.51 vs. 0.25 ± 0.43). This result is consistent with a previous study in which patients were treated with these same three oral BPs. Paggiosi et al. conducted a two-year, open-label, randomized controlled trial of the three oral bisphosphonates in postmenopausal osteoporosis [25]. In that TRIO study, patients were randomized into three groups: (i) ibandronate, 150 mg/month; (ii) alendronate, 70 mg/week; or (iii) risedronate, 35 mg/week. All received calcium (1200 mg/day) and vitamin D (800 IU/day) for two years. In the ibandronate and alendronate groups, BMD increases in the lumbar spine and total body were greater than those in the risedronate group (6.68% vs. 6.84% vs. 3.04% for the lumbar spine; 1.91% vs. 2.19% vs. 1.12% for the total body).

In this study, the increase in total BMD of ibandronate was better than that in the risedronate group in patients with comorbid osteoporosis and LSS. A previous study also reported the greater efficacy of ibandronate for osteoporosis than that of risedronate. Using a database from the Korean National Health Insurance Service, Lee et al. compared the fracture prevention efficacy of ibandronate taken once a month (150 mg) and risedronate (150 mg) taken once a month for Korean women aged 60 years or older from 2006 to 2015 [26]. After propensity score matching for 36,701 patients using once-monthly ibandronate or risedronate, data from 3454 patients in each group were compared. After four years of follow-up, the ibandronate group had significantly lower incidences of non-vertebral and overall fractures than those of the risedronate group [incidence rate ratio (IRR) = 0.798, 95% CI 0.647–0.985, *p* = 0.036 and IRR = 0.822, 95% CI 0.698–0.968, *p* = 0.019, respectively]. In terms of compliance, ibandronate may be better than risedronate. Chung et al. reported that postmenopausal osteoporotic women in Korea strongly preferred monthly ibandronate over weekly risedronate due to convenience [27]. In the monthly ibandronate group, gastrointestinal side effects such as nausea and abdominal distension were fewer than those in the weekly risedronate group.

This study has some limitations. First, objective measures for walking time and bone turnover markers were not assessed because of the retrospective study design. Questionnaires on daily activities or bone turnover markers can help identify the mechanism. Second, the compliance of patients was not evaluated despite reported low compliance with oral bisphosphonates. However, compliance was not expected to be low because we included patients with a follow-up period of more than three years. Third, male patients were not included in this study because ibandronate is only allowed for female patients. In addition, a consistent protocol for which to use bisphosphonate was not applied. Finally, we evaluated only the effect of osteoporosis treatment, not the effect of fracture prevention. Further studies are needed to extend the results into other types of osteoporosis medication, including denosumab and bone-forming agents such as romosozumab and teriparatide. It is also necessary to study how active treatment for LSS, such as spinal surgery, helps treat osteoporosis. Despite these limitations, this study is the first to evaluate the effect of LSS on BMD in patients receiving one of three oral bisphosphonates (ibandronate, alendronate and risedronate) for newly diagnosed osteoporosis.

## 5. Conclusions

Symptomatic LSS interfered with treatment in patients with newly diagnosed osteoporosis. Ibandronate and alendronate were more effective in treating osteoporosis than risedronate. In particular, ibandronate was more effective than risedronate in patients with comorbid osteoporosis and LSS. Therefore, to increase BMD in patients with both osteoporosis and LSS, active treatment for LSS is needed, and ibandronate may be the preferred treatment among the three BPs. However, further trials are needed to validate the results of this study and extend them to other drugs.

## Figures and Tables

**Figure 1 jcm-12-02027-f001:**
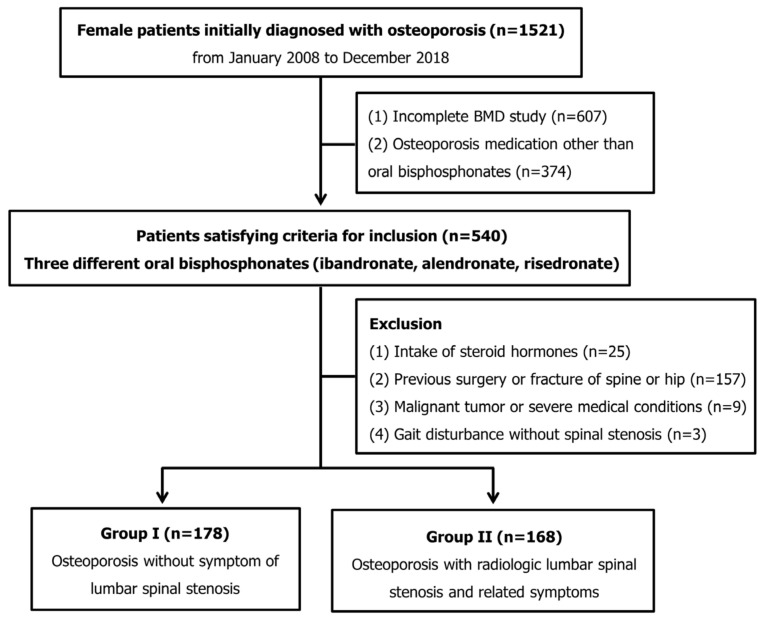
Flow chart of patient inclusion in this study.

**Figure 2 jcm-12-02027-f002:**
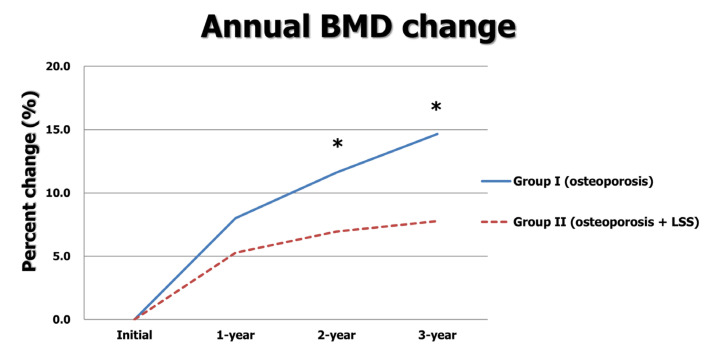
Annual BMD changes between two groups. * indicates *p* < 0.05.

**Figure 3 jcm-12-02027-f003:**
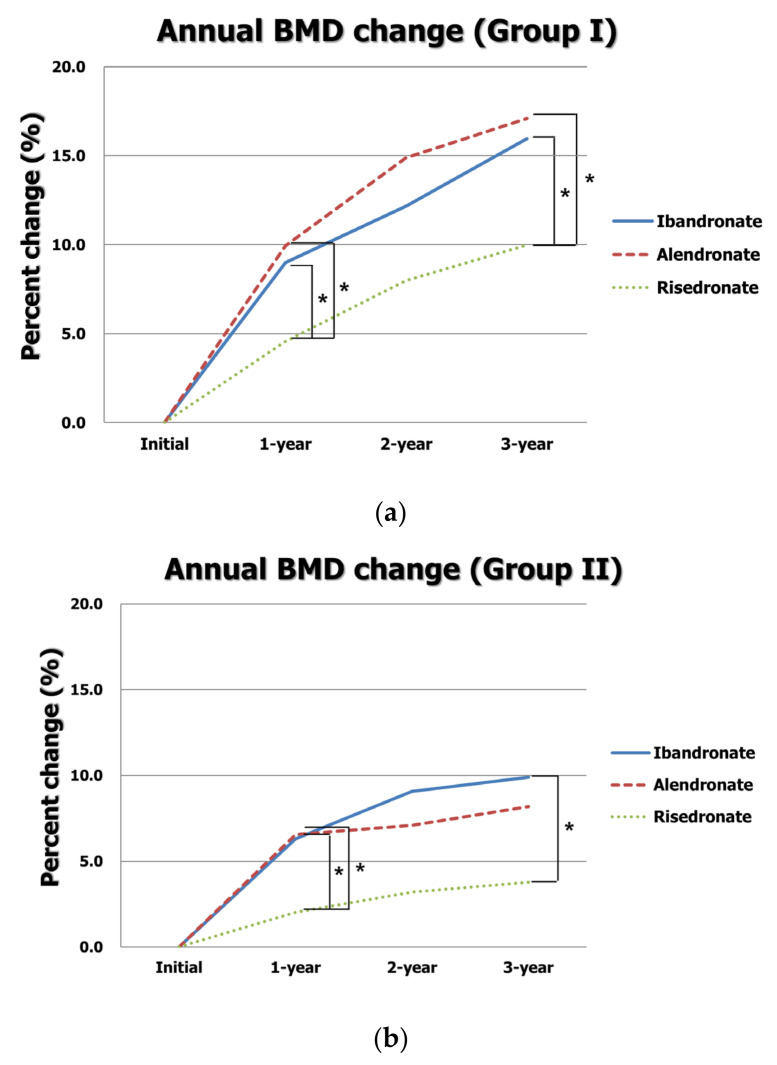
Annual changes in BMD according to type of BP in (**a**) group I and (**b**) group II. * indicates *p* < 0.05 in a post hoc analysis.

**Table 1 jcm-12-02027-t001:** Patient characteristics and BMD T-scores between the two groups.

Parameters	Group I (Osteoporosis)	Group II (Osteoporosis + LSS)	*p*-Value
Patient number	178	168	
Female:male	178:0	168:0	1.000
Age (years)	74.8 ± 7.9	74.6 ± 8.3	0.834
BMI (kg/m^2^)	23.7 ± 3.1	23.2 ± 3.0	0.401
Ibandronate:alendronate:risedronate	76:49:53	66:55:47	0.571
BMD T-score			
Initial	−3.62 ± 0.50	−3.60 ± 0.63	0.620
1-year F/U	−3.34 ± 0.47	−3.40 ± 0.49	0.248
2-year F/U	−3.17 ± 0.66	−3.36 ± 0.61	0.006
3-year F/U	−3.09 ± 0.49	−3.31 ± 0.62	<0.001
Annual change in BMD			
1-year F/U	0.29 ± 0.32	0.19 ± 0.33	0.007
2-year F/U	0.13 ± 0.29	0.06 ± 0.36	0.028
3-year F/U	0.11 ± 0.28	0.04 ± 0.33	0.026
Total change in BMD	0.53 ± 0.47	0.28 ± 0.45	<0.001

Note: BMD—bone mineral density; LSS—lumbar spinal stenosis; BMI—body mass index; F/U—follow-up.

**Table 2 jcm-12-02027-t002:** Annual BMD T-scores and changes in BMD according to the three different types of BP.

	Ibandronate (I)	Alendronate (A)	Risedronate (R)	*p*-Value Post hoc Test
Patient number	142	104	100	
Female:male	142:0	104:0	100:0	1.000
Age (years)	74.6 ± 8.5	75.1 ± 8.4	74.6 ± 7.2	0.829
BMI (kg/m^2^)	23.3 ± 2.8	23.4 ± 3.2	23.6 ± 3.2	0.885
BMD T-score				
Initial	−3.67 ± 0.54	−3.64 ± 0.58	−3.48 ± 0.57	0.022 I vs. R = 0.025
1-year F/U	−3.38 ± 0.48	−3.36 ± 0.55	−3.36 ± 0.0.57	0.913
2-year F/U	−3.30 ± 0.54	−3.25 ± 0.56	−3.22 ± 0.83	0.668
3-year F/U	−3.19 ± 0.55	−3.19 ± 0.57	−3.23 ± 0.57	0.858
Annual change in BMD				
1-year F/U	0.29 ± 0.31	0.29 ± 0.33	0.12 ± 0.33	<0.001 I vs. R < 0.001 A vs. R = 0.001
2-year F/U	0.10 ± 0.38	0.10 ± 0.33	0.08 ± 0.24	0.832
3-year F/U	0.09 ± 0.34	0.06 ± 0.32	0.06 ± 0.24	0.588
Total change in BMD	0.49 ± 0.46	0.45 ± 0.51	0.25 ± 0.43	<0.001 I vs. R < 0.001 A vs. R = 0.004

Note: BMD—bone mineral density; LSS—lumbar spinal stenosis; BMI—body mass index; F/U—follow-up.

**Table 3 jcm-12-02027-t003:** Annual BMD T-scores and changes in BMD according to the three different types of BP in each group.

Parameters	Group I (Osteoporosis, n = 178)				Group II (Osteoporosis + LSS, n = 168)			
	Ibandronate (I)	Alendronate (A)	Risedronate (R)	*p*-Value Post hoc Test	Ibandronate (I)	Alendronate (A)	Risedronate (R)	*p*-Value Post hoc Test
Patient number	76	49	53		66	55	47	
Female:male	76:0	49:0	53:0	1.000	66:0	55:0	47:0	1.000
Age (years)	75.0 ± 8.4	74.7 ± 8.4	74.6 ± 6.9	0.958	74.0 ± 8.7	75.5 ± 8.4	74.5 ± 7.6	0.606
BMI (kg/m^2^)	23.7 ± 3.3	23.9 ± 2.9	23.6 ± 3.3	0.944	23.1 ± 2.7	22.8 ± 3.5	23.9 ± 3.3	0.683
BMD T-score								
Initial	−3.70 ± 0.46	−3.63 ± 0.51	−3.51 ± 0.53	0.099	−3.64 ± 0.62	−3.66 ± 0.64	−3.44 ± 0.61	0.170
1-year F/U	−3.36 ± 0.40	−3.29 ± 0.50	−3.35 ± 0.52	0.694	−3.41 ± 0.56	−3.42 ± 0.60	−3.37 ± 0.63	0.921
2-year F/U	−3.25 ± 0.46	−3.09 ± 0.42	−3.12 ± 0.99	0.338	−3.34 ± 0.62	−3.40 ± 0.63	−3.33 ± 0.59	0.860
3-year F/U	−3.11 ± 0.43	−3.01 ± 0.51	−3.16 ± 0.55	0.286	−3.33 ± 0.65	−3.35 ± 0.63	−3.30 ± 0.58	0.820
Annual change in BMD								
1-year F/U	0.34 ± 0.28	0.34 ± 0.32	0.16 ± 0.35	0.002 I vs. R = 0.004 A vs. R = 0.009	0.23 ± 0.33	0.24 ± 0.33	0.07 ± 0.31	0.017 I vs. R = 0.031 A vs. R = 0.031
2-year F/U	0.11 ± 0.28	0.20 ± 0.35	0.12 ± 0.24	0.205	0.10 ± 0.46	0.02 ± 0.28	0.04 ± 0.24	0.476
3-year F/U	0.15 ± 0.25	0.09 ± 0.35	0.08 ± 0.26	0.311	0.03 ± 0.41	0.04 ± 0.30	0.03 ± 0.22	0.983
Total change in BMD	0.59 ± 0.40	0.62 ± 0.53	0.35 ± 0.46	0.003 I vs. R = 0.007 A vs. R = 0.015	0.36 ± 0.49	0.30 ± 0.45	0.13 ± 0.38	0.021 I vs. R = 0.018

Note: BMD—bone mineral density; LSS—lumbar spinal stenosis; BMI—body mass index; F/U—follow-up.

## Data Availability

Original data will be made available upon reasonable request.

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
