# Peer review of "Effect of Lumbar Spinal Stenosis on Treatment of Osteoporosis: Comparison of Three Oral Bisphosphonate Therapies"

_jcm, 2023, doi:10.3390/jcm12052027_

Round 1
Reviewer 1 Report
This is a retrospective analysis evaluation the effect LSS on bone density increase in women with osteoporosis that are treated with BPs.
there are two major questions regarding the methodology:
1- The authors shoed association between LSS bone density increase in women with osteoporosis that are treated with BPs. Association does not mean causation. In table 1 the author showed few demographic variables that were comparable between the groups. However, this doesn't guarantee the lack of confounders. Other variables should have been taken into account such as other comorbidities and medications that can interfere with BPs effect on bone density. Generally speaking, In order to show that the differences in the bone density increase were the result of LSS presence multivariate regression analysis is needed
2-The authors did not give any possible explanation for their results . Why the authors think that LSS effect bone density increase in woman treated with BPS
In addition the introduction is some how confusing. The description and the background of LSS and osteoporosis are too long. lines 44-94 - the presence of pulse is not a neurological symptom. The introduction shoed be revised and focused on the study question and rational
Author Response
Response to Reviewer 1 Comments
Point 1: The authors shoed association between LSS bone density increase in women with osteoporosis that are treated with BPs. Association does not mean causation. In table 1 the author showed few demographic variables that were comparable between the groups. However, this doesn't guarantee the lack of confounders. Other variables should have been taken into account such as other comorbidities and medications that can interfere with BPs effect on bone density. Generally speaking, In order to show that the differences in the bone density increase were the result of LSS presence multivariate regression analysis is needed.
Response 1: Thank you for your comment. We agree with your comment. Our previous study evaluated the effect of lumbar spinal stenosis (LSS) on bone mineral density (BMD) in patients undergoing osteoporosis treatment with ibandronate. We demonstrated that symptomatic LSS was the only independent risk factor for continuous BMD improvement (odds ratio = 0.316) in multivariate analysis. In this study, we aimed to extend the results and mainly evaluate the therapeutic efficacies of bisphosphonates (BPs) in patients treated with one of three oral BPs (ibandronate, alendronate, and risedronate). We added the previous results regarding multivariate analysis according to your comment.
Park, H.Y.; Ha, J.Y.; Kim, K.W.; Baek, I.H.; Park, S.B.; Lee, J.S. Effect of lumbar spinal stenosis on bone mineral density in osteoporosis patients treated with ibandronate. BMC Musculoskelet Disord 2021, 22, 412.
Point 2: The authors did not give any possible explanation for their results . Why the authors think that LSS effect bone density increase in woman treated with BPS.
Response 2: Thank you for your comment. We have suggested several mechanisms in our previous study. LSS causes neurologic claudication and reduces the strength of the lower limb, which decreases physical activity. Walking difficulty due to claudication or physical inactivity can be associated with decreased BMD. In patients with vascular claudication originating from peripheral arterial disease, a relationship between physical inactivity and decreased BMD was also reported. In this regard, physical activities in seniors can induce the maintenance of BMD or increase BMD through the physical load. Lee et al. have also reported that increased physical activity and regular walking exercise could prevent osteoporosis in a study of older women aged 65 years and over. We added the mechanism to the discussion according to your comment.
Lee I, Ha C, Kang H. Association of sarcopenia and physical activity with femur bone mineral density in elderly women. J Exerc Nutrition Biochem. 2016;20:23-8.
Point 3: In addition the introduction is some how confusing. The description and the background of LSS and osteoporosis are too long. lines 44-94 - the presence of pulse is not a neurological symptom. The introduction shoed be revised and focused on the study question and rational
Response 3: Thank you for your comment. We have reduced and revised the introduction part based on important research questions and grounds according to your opinion.

Reviewer 2 Report
The idea of the study is interesting. I have to admit that I am mainly interested in the results that were already described in the author’s previous article. They state in the discussion: ‘In our previous study, we demonstrated that BMD increases were lower in patients with both osteoporosis and LSS than in patients with osteoporosis alone during treatment with ibandronate [5].’
I was somewhat critical at first, because patients with the most clinically relevant neurogenic claudication (the ones that are operated on) are excluded. However, the results prove that there is a difference between the groups with and without LSS, indicating that the decrease in movement does make a difference. There is however no measure to quantify the difference in walking time between the two groups. A critical evaluation concerning which other factors could have influenced the properties of both groups should be added to the discussion. Beware of the following: The authors state that as a treatment for LSS injections with corticosteroids are being administered. Patients that orally take corticosteroids are excluded, but no further information is given on the frequency or dosage of the cortico injections. If one is critical one might remark that the corticosteroid injections in the LSS group cause them to have a worse outcome in osteoporosis outcome. Please give info on the cortico injections and make remarks on this in the discussion.
In the current study the distinction is being made in type of medication to improve osteoporosis. My question concerns the decision making in what medication was prescribed. I could not retrieve information about that. What did it depend on? How was it decided which patient got which regimen? Furthermore, it is stated that risedronate could be administered weekly orally or by injection monthly. This is the only medicine in which this choice is given and the conclusion is that medicine has the worst results. That makes it extra important to be clear about the administration. Patients compliance for taking the medication is best evaluated with injections.
Finally, the article is very well written and reads easily.
Author Response
Response to Reviewer 2 Comments
Point 1: The idea of the study is interesting. I have to admit that I am mainly interested in the results that were already described in the author’s previous article. They state in the discussion: ‘In our previous study, we demonstrated that BMD increases were lower in patients with both osteoporosis and LSS than in patients with osteoporosis alone during treatment with ibandronate [5].’
I was somewhat critical at first, because patients with the most clinically relevant neurogenic claudication (the ones that are operated on) are excluded. However, the results prove that there is a difference between the groups with and without LSS, indicating that the decrease in movement does make a difference. There is however no measure to quantify the difference in walking time between the two groups. A critical evaluation concerning which other factors could have influenced the properties of both groups should be added to the discussion. Beware of the following: The authors state that as a treatment for LSS injections with corticosteroids are being administered. Patients that orally take corticosteroids are excluded, but no further information is given on the frequency or dosage of the cortico injections. If one is critical one might remark that the corticosteroid injections in the LSS group cause them to have a worse outcome in osteoporosis outcome. Please give info on the cortico injections and make remarks on this in the discussion.
Response 1: We totally agree with your comment. Due to the retrospective study design, lack of objective measure for walking time or dailiy activities is a major limitation of our study as your comment. We added this in the limitation section. We excluded the patients with continuous use of steroid hormones for the treatment of diseases such as rheumatoid arthritis or chronic lung disease. Steroid injections were performed only three or less times using 5mg of dexamethasone. This amount of steroid is not thought to affect the progression of osteoporosis. We revised the manuscript including this according to your comment.
Point 2: In the current study the distinction is being made in type of medication to improve osteoporosis. My question concerns the decision making in what medication was prescribed. I could not retrieve information about that. What did it depend on? How was it decided which patient got which regimen? Furthermore, it is stated that risedronate could be administered weekly orally or by injection monthly. This is the only medicine in which this choice is given and the conclusion is that medicine has the worst results. That makes it extra important to be clear about the administration. Patients compliance for taking the medication is best evaluated with injections.
Response 2: Thank you for your comment. In this study, there was no protocol for the drug selection process, and different doctors usually prescribed different bisphosphonates. We compared the pooled effects of different bisphosphonate drugs retrospectively. It was added to the limitation that there was no consistent protocol for drug selection. The sentence to administration of risedronate injections was miswritten while writing the manuscript referring to other previous studies. The patients treated with oral bisphosphonates were included, and we deleted the related sentence from the manuscript.
Point 3: Finally, the article is very well written and reads easily.
Response 3: Thank you very much for the positive comment.

Reviewer 3 Report
1. In line 9:. All patients with LSS were treated conservatively with medications and epidural steroid injections. This contradicts what is stated on line 73: Exclusion criteria were (1) use of steroid hormones
2. In lines 105 and 106 : “Discordant BMD values of the spine with aortic calcification and lumbar spondylosis were excluded”. But there no information about this exclusion in Figure 1 and in exclusion criteria .
3. Indicate the registration documents of equipments which were used in densitometry.
4. In line 105: “Discordant BMD values of the spine” You mean “Discordance BMD between the Lumbar Spine and Femoral Neck” Please specify.
5. You used T-score results in BMD study. However, T-score and BMD are not the same. It is better to indicate BMD in results as "BMD T -score' or shortened otherwise.
6. Diagram 2 and Table 1 show the same data. maybe it's better to specify other indicators as percentage improvement or something else?
7. Diagram 1 and Table 1 show the same data. maybe it's better to specify other indicators as percentage improvement or something else?
8. Diagram 3 and Table 3 show the same data. maybe it's better to specify other indicators as percentage improvement or something else?.
9. In table 1: There are abbreviated terms BMD, bone mineral density; LSS, lumbar spinal stenosis; BMI, body mass index; F/U, follow-up. It is necessary to indicate at the bottom of the table after the word Note:
10. At the bottom of table 2 and 3, abbreviated terms must be indicated as follows: Note: BMD - bone mineral density; LSS - lumbar spinal stenosis; BMI - body mass index; F/U - follow-up.
11. The text notes a comparison between the treatment outcomes of ibandoronate and alendronate versus risedronate. It would be better if a comparison was made between ibandronate and alendronate at the beginning and prove that there are no significant differences between them. then we obtain the mean values ​​of the treatment results between the mean values ​​of ibandronate and alendronate and compare them with the results of treatment with risedronate.
12. On fig. 3 there are two diagrams (A) and (B). Please change the capital letter to small letter (a) and (b). The link to the figure should be without specifying a part of the figure 3 as figure 3a or 3b.
13. This study complies with the Declaration of Helsinki and was performed according to ethics committee approval.
14. Must be noted that this study had institutional review board approval, and the need to obtain informed patient consent was waived, institutional review board approval, and all patients provided written informed consent. Must be there written informed consent was obtained from all patients, and the study protocol was approved by the institutional committee on human research, ensuring that it conformed to the ethical guidelines of the 1975 Declaration of Helsinki.
Author Response
Response to Reviewer 3 Comments
Point 1: In line 9:. All patients with LSS were treated conservatively with medications and epidural steroid injections. This contradicts what is stated on line 73: Exclusion criteria were (1) use of steroid hormones
Response 1: Thank you for your comment. We excluded the patients with continuous use of steroid hormones for the treatment of diseases such as rheumatoid arthritis or chronic lung disease. Steroid injections were performed only three or fewer times using 5mg of dexamethasone. This amount of steroid is not thought to affect the progression of osteoporosis. We revised the sentence including this according to your comment.
Point 2: In lines 105 and 106 : “Discordant BMD values of the spine with aortic calcification and lumbar spondylosis were excluded”. But there no information about this exclusion in Figure 1 and in exclusion criteria .
Response 2: Thank you for your comment. According to a previous study, discordant BMD values of the spine were defined as a discrepancy of >1 standard deviation in T-score between adjacent vertebrae. We added this sentence reflecting your comment.
Point 3: Indicate the registration documents of equipments which were used in densitometry.
Response 3: Thank you for your comment. We used the Lunar Prodigy for dual-energy X-ray absorptiometry. We suggested the name of the equipment (Lunar Prodigy; GE Healthcare Bio-Sciences Corp., Piscataway, NJ, USA) in the BMD measurement section.
Point 4: In line 105: “Discordant BMD values of the spine” You mean “Discordance BMD between the Lumbar Spine and Femoral Neck” Please specify.
Response 4: Thank you for your comment. As previously mentioned, discordant BMD values of the spine were defined as a discrepancy of >1 standard deviation in T-score between adjacent vertebrae. We added this sentence reflecting your comment.
Point 5: You used T-score results in BMD study. However, T-score and BMD are not the same. It is better to indicate BMD in results as "BMD T -score' or shortened otherwise.
Response 5: We totally agree with your comment. BMD of methods and results section was modified to BMD T-score according to your comment.
Point 6: Diagram 2 and Table 1 show the same data. maybe it's better to specify other indicators as percentage improvement or something else?
Response 6: Thank you for your comment. We agree with you. According to your comment, we revised Figure 2 with a percent change as an indicator.
Point 7: Diagram 1 and Table 1 show the same data. maybe it's better to specify other indicators as percentage improvement or something else?
Response 7: Thank you for your comment. Figure 1 is the flow chart of patient inclusion in this study. Maybe what you pointed out is in figure 2 and figure 3.
Point 8: Diagram 3 and Table 3 show the same data. maybe it's better to specify other indicators as percentage improvement or something else?
Response 8: Thank you for your comment. We agree with you. We revised Figure 3a and 3b based on the percent change according to your comment.
Point 9: In table 1: There are abbreviated terms BMD, bone mineral density; LSS, lumbar spinal stenosis; BMI, body mass index; F/U, follow-up. It is necessary to indicate at the bottom of the table after the word Note:
Response 9: Thank you for your comment. We corrected the abbreviated terms according to your comment.
Point 10: At the bottom of table 2 and 3, abbreviated terms must be indicated as follows: Note: BMD - bone mineral density; LSS - lumbar spinal stenosis; BMI - body mass index; F/U - follow-up.
Response 10: Thank you for your comment. We corrected the abbreviated terms according to your comment.
Point 11: The text notes a comparison between the treatment outcomes of ibandoronate and alendronate versus risedronate. It would be better if a comparison was made between ibandronate and alendronate at the beginning and prove that there are no significant differences between them. then we obtain the mean values ​​of the treatment results between the mean values ​​of ibandronate and alendronate and compare them with the results of treatment with risedronate.
Response 11: Thank you for your comment. We revised the comparison of drugs in the discussion section according to your comment.
Point 12: On fig. 3 there are two diagrams (A) and (B). Please change the capital letter to small letter (a) and (b). The link to the figure should be without specifying a part of the figure 3 as figure 3a or 3b.
Response 12: Thank you for your comment. We modified the capital letter to small letter according to your comment.
Point 13: This study complies with the Declaration of Helsinki and was performed according to ethics committee approval.
Response 13: Thank you for your comment. We added this sentence according to your comment.
Point 14: Must be noted that this study had institutional review board approval, and the need to obtain informed patient consent was waived, institutional review board approval, and all patients provided written informed consent. Must be there written informed consent was obtained from all patients, and the study protocol was approved by the institutional committee on human research, ensuring that it conformed to the ethical guidelines of the 1975 Declaration of Helsinki.
Response 14: Thank you for your comment. We added this sentence according to your comment.
